

# HMCFormer (hierarchical multi-scale convolutional transformer): a hybrid CNN+Transformer network for intelligent VIA screening

Bo Feng[1,2], Chao Xu[1,2], Zhengping Li[1,2] and Chuanyi Zhang[3]

[1] School of Integrated Ciruits, Anhui University, HeFei, Anhui, China
[2] Anhui Engineering Laboratory of Agro-Ecological Big Data, Anhui University, HeFei, Anhui, China
[3] Obstetrics and Gynecology, People's Hospital of Fanchang District, Wuhu, Anhui, China

Corresponding authors
Chao Xu, xchao@ahu.edu.cn
Zhengping Li, 04173@ahu.edu.cn

## ABSTRACT

Cervical cancer ranks first in incidence among malignant tumors of the female reproductive system, and 80% of women who die from cervical cancer worldwide are from developing countries. Visual inspection with acetic acid (VIA) screening based on artificial intelligence-assisted diagnosis can provide a cheap and rapid screening method. This will attract more low-income women to volunteer for regular cervical cancer screening. However, current AI-based VIA screening studies either have low accuracy or require expensive equipment assistance. In this article, we propose the Hierarchical Multi-Scale Convolutional Transformer network, which combines the hierarchical feature extraction capability of Convolutional Neural Network (CNNs) and the global dependency modeling capability of Transformers to address the challenges of realizing intelligent VIA screening. Hierarchical multi-scale convolutional transformer (HMCFormer) can be divided into a Transformer branch and a CNN branch. The Transformer branch receives unenhanced lesion sample images, and the CNN branch receives lesion sample images enhanced by the proposed dual-color space-based image enhancement algorithm. The authors design a hierarchical multi-scale pixel excitation module for adaptive multi-scale and multi-level local feature extraction. The authors apply the structure of the Swin Transformer network with minor modifications in the global perception modeling process. In addition, the authors propose two feature fusion concepts: adaptive preprocessing and superiority-inferiority fusion, and design a feature fusion module based on these concepts, which significantly improves the collaborative ability of the Transformer branch and the CNN branch. The authors collected and summarized 5,000 samples suitable for VIA screening methods from public datasets provided by companies such as Intel and Google, forming the PCC5000 dataset. On this dataset, the proposed algorithm achieves a screening accuracy of 97.4% and a grading accuracy of 94.8%.

## INTRODUCTION

Cervical cancer, with the highest incidence rate among malignant tumors of the female reproductive system, is witnessing an upward trend globally. According to data from the World Health Organization (WHO), there are approximately 570,000 new cases of cervical cancer worldwide each year, resulting in 280,000 deaths, which account for 7.5% of all cancer-related deaths. In developing countries, cervical cancer is the most prevalent type of cancer among women, particularly in regions such as Africa, Latin America, and Asia (*Qiu et al., 2024*). According to the 2023 statistics released by the National Cancer Center of China, there were 119,300 new cases of cervical cancer and 37,200 deaths in China (*Qiu et al., 2024*).

Currently, the commonly used cervical cancer screening methods include ThinPrep Cytology Test (TCT) screening, human papillomavirus (HPV) screening, TCT+HPV, and visual inspection with acetic acid (VIA)/Visual Inspection with Lugol's Iodine (VILI) screening. TCT and HPV screenings are the most widely used early screening methods for cervical cancer. TCT screening is a method where a cervical brush or spatula is used to collect cell samples from the surface of the cervix, which are then sent to a laboratory for processing and microscopic analysis to determine the presence of cervical abnormalities. HPV screening, on the other hand, is a molecular-level test that analyzes genetic material (DNA or RNA) to detect the presence of high-risk HPV infections. The combination of TCT and HPV screening, known as TCT+HPV screening, integrates both methods and is currently the most widely used approach for cervical cancer screening. This combined method has relatively high sensitivity and specificity, allowing for more effective early detection and treatment of cervical cancer. However, it is more costly and time-consuming compared to other methods (*Qureshi, Das & Zahra, 2010*). VIA/VILI screening involves visual inspection with acetic acid and Lugol's iodine staining. Similar to cytological examinations, this method is effective for cervical cancer screening. It involves applying acetic acid and iodine to the cervix and visually inspecting it to determine the presence and grade of lesions. This method is cost-effective, simple to operate, and provides timely diagnosis (*Qureshi, Das & Zahra, 2010*; *Sankaranarayanan et al., 2004*; *Ferreccio et al., 2003*), making it particularly suitable for economically underdeveloped areas. Due to the large number of patients and the limited number of skilled colposcopists, the accuracy of colposcopy in resource-limited areas is relatively low (*Khan, Werner & Darragh, 2017*). *Underwood et al. (2012)* reported that in some low- and middle-income countries, the average positive detection rate of colposcopic biopsy is 63.3%, with frequent occurrences of over-diagnosis or under-diagnosis.

In summary, developing AI-assisted VIA screening methods is highly necessary. AI algorithms can effectively address the issue of the limited number of colposcopists and significantly reduce screening costs. This will encourage more low-income women to voluntarily undergo regular cervical cancer screenings.

Currently, AI-based TCT+HPV screening has become quite mature and has been widely applied in actual screening practices (*Zhang et al., 2017*; *Rahaman et al., 2021*; *Singh & Goyal, 2021*; *Holmström et al., 2020*; *Tang et al., 2021*). However, despite their good

performance in terms of accuracy, these technologies have not shown significant effects in reducing the time costs of cervical cancer screening, nor have they substantially reduced the screening costs borne by patients. Nonetheless, these AI-based screening technologies can bring a certain degree of cost savings and improved work efficiency to healthcare institutions, but these benefits do not directly translate into advantages for patients.

In contrast, AI-assisted VIA screening methods still require further development. Using SVM (*Asiedu et al., 2018*), 134 cervical tissue samples were classified into normal and abnormal categories, achieving an accuracy of 80%. Another study (*Miyagi, Takehara & Miyake, 2019*) classified 310 cervical images for CIN classification, achieving an accuracy of 82.3%. Using FR-CNN for detecting precancerous cervical cells (*Hu et al., 2019*) with 279 images showed an accuracy of 89%. However, these methods used a small number of samples, which are not representative. A multimodal CNN trained with 60,000 colposcopy images and clinical inferences from physicians (*Song et al., 2014*) achieved an accuracy of 89%, but this method only performed binary classification and could not distinguish the grades of cervical intraepithelial neoplasia (CIN). *Zhang et al. (2020)* used DenseNet CNN to identify cervical intraepithelial neoplasia, but the accuracy was only 73% without the interference of other cervical disease samples. A recent deep learning technique, ColpoNet, developed from the pretrained DenseNet model (*Saini et al., 2020*), performed multiclass classification on four types of cervical cancer and achieved an accuracy of 81.3%. *Li et al. (2020)* proposed a method for "computer-assisted cervical cancer diagnosis using time-lapse colposcopic images," where five images were taken every 30 s after applying acetic acid to capture the features of cervical intraepithelial neoplasia for AI recognition. However, this method requires filtered colposcopes, which are expensive and not suitable for promotion in economically underdeveloped areas. Additionally, taking five delayed images significantly alters the traditional operation procedures of physicians, and its accuracy is only 79%. *Luo et al. (2020)* proposed a deep learning-based method that combines multiple decision features of CNNs for the classification and diagnosis of cervical lesions. The classification accuracy rate for 600 cervical images was only 83.5% at most. *Fang et al. (2022)* proposed a deep reverse residual network based on an improved channel attention mechanism for CIN grading. A total of 6,996 samples were used, and the grading accuracy was 81.38%.

In summary, current AI-assisted diagnostic VIA screening methods are mainly based on CNN networks, but their accuracy has not yet reached a practical level. With the introduction of the Vision Transformer (ViT) network, Transformer networks have gained widespread popularity. Although Transformer networks have strong capabilities in global context feature extraction, their ability to extract detailed local features is relatively weak, which is why they are seldom used in medical image processing. To combine the advantages of Transformer networks in global information extraction with the strengths of CNNs in local feature extraction, CNN+Transformer fusion networks have been proposed. CNN+Transformer networks have also been widely applied in medical image processing (*Xie et al., 2021*; *Shao et al., 2021*; *Li et al., 2024*).

This article proposes a CNN+Transformer fusion network called hierarchical multi-scale convolutional transformer (HMCFormer) for intelligent VIA screening. The

neural network framework of HMCFormer effectively leverages the collaborative strengths of CNNs and Transformers, demonstrating strong low-level feature extraction capabilities as well as good performance in capturing global contextual features.

This study presents several notable innovations:

(1) A dual-color space enhancement and fusion algorithm is proposed, which decomposes images into the Lab and YCrCb color spaces. By increasing the color difference between the lesion area and the normal cervical skin, the lesion area is accentuated.

(2) The Hierarchical Multi-Scale Pixel Excitation (HMSPE) module is proposed, which integrates multi-scale feature extraction and multi-level feature extraction with minimal computational overhead.

(3) Currently, there is no evaluation metric to demonstrate that the CNN+Transformer fusion network fully maximizes its collaborative potential. To address this, we propose using mutual information from information theory to evaluate different fusion methods. We find that performing adaptive preprocessing on the CNN and Transformer branches before fusion, and incorporating an "advantage and disadvantage fusion" strategy during fusion, can better leverage the collaborative potential of the parallel CNN+Transformer network.

(4) In this article, the unaugmented images are fed into the Transformer branch, while the augmented images are fed into the CNN branch. This approach improves the network's recognition accuracy and eliminates the impact of image augmentation on lesion grading.

(5) After organizing and summarizing multiple public datasets, 5,000 samples suitable for VIA screening were obtained, forming the PCC5000 dataset.

## RELATED WORK

In this section, we will first briefly explain the VIA/VILI screening process and the criteria for grading cervical cancer. Finally, we will provide a brief introduction to the CNN+Transformer network and propose that moderate mutual information is more suitable for the CNN+Transformer network.

### VIA screening procedure and cervical cancer staging criteria

The screening process begins with using a dry cotton ball to wipe off any acetic acid and mucus, followed by a gentle swabbing of the cervix with a saline-soaked cotton ball or a dry cotton swab. Subsequently, the entire cervix is coated with a 5% acetic acid solution using a cotton ball. After a 1-min interval, the reaction of the cervical epithelium to the acetic acid is directly observed under standard illumination. The lesion area presents as white. A preliminary diagnosis is made, and the results are documented based on the thickness, border, and contour of the white lesions (*Liu, 2014*). As shown in Fig. 1.

The cervical cancer is staged as follows:

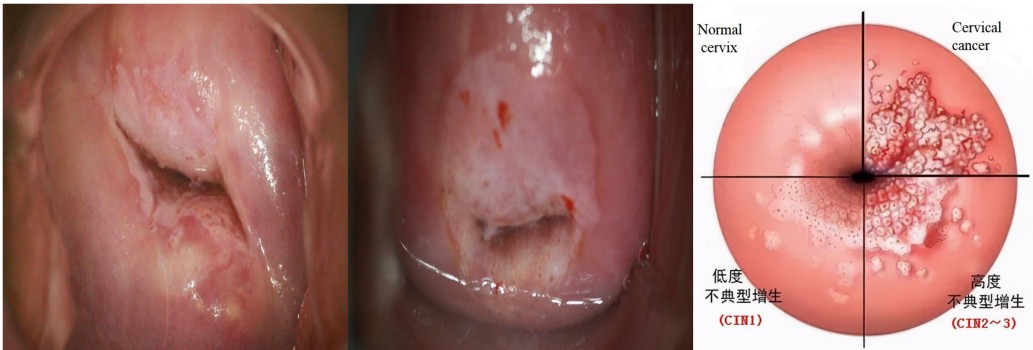

**Figure 1 Examples of cervical intraepithelial neoplasia.**

(A) Normal reaction: Healthy cervical cells typically do not undergo any changes post-acetic acid application. The color of the cervix usually remains unchanged or becomes slightly paler.

(B) CIN1: Post-acetic acid application, there may be a mild white change, which might not be prominent and only involves the basal third of the cervical epithelium.

(C) CIN2: Post-acetic acid application, the lesion area typically turns white, which is more pronounced than in CIN1. These white areas involve approximately two-thirds of the squamous epithelium.

(D) CIN3: Post-acetic acid application, the lesion area prominently turns white. These white areas involve the upper layer or the entire layer of the squamous epithelium, manifesting as distinct irregular, coarse, yellow-white lesions.

## Comparison of CNN and transformer fusion types

CNNs are particularly adept at capturing local features such as edges, corners, and textures. However, standalone CNNs often have limitations when it comes to handling global information. Transformer networks can model long-range dependencies and achieve global receptive field coverage, but they lack the inductive bias capabilities of CNNs (*Alrfou, Zhao & Kordijazi, 2023*; *d'Ascoli et al., 2021*). In the medical field, the fusion of CNN and Transformer networks has gained widespread application (*Xie et al., 2021*; *Shao et al., 2021*; *Li et al., 2024*). In medical image recognition, lesions require sensitivity to both local texture features and strong global contextual information processing capabilities. Given that the number of samples is usually limited, the fusion of CNN and Transformer networks is highly suitable.

Currently, CNN and Transformer fusion methods can be roughly divided into three types: early layer fusion, sequential fusion, and parallel fusion. Given the distinct differences in feature extraction and computational mechanisms between Transformers and CNNs, parallel fusion networks divide the entire network into a Transformer branch and a CNN branch. These two branches operate independently while sharing information with each other. This allows the entire network to leverage the strengths of both network structures while mitigating their respective weaknesses. There are many successful

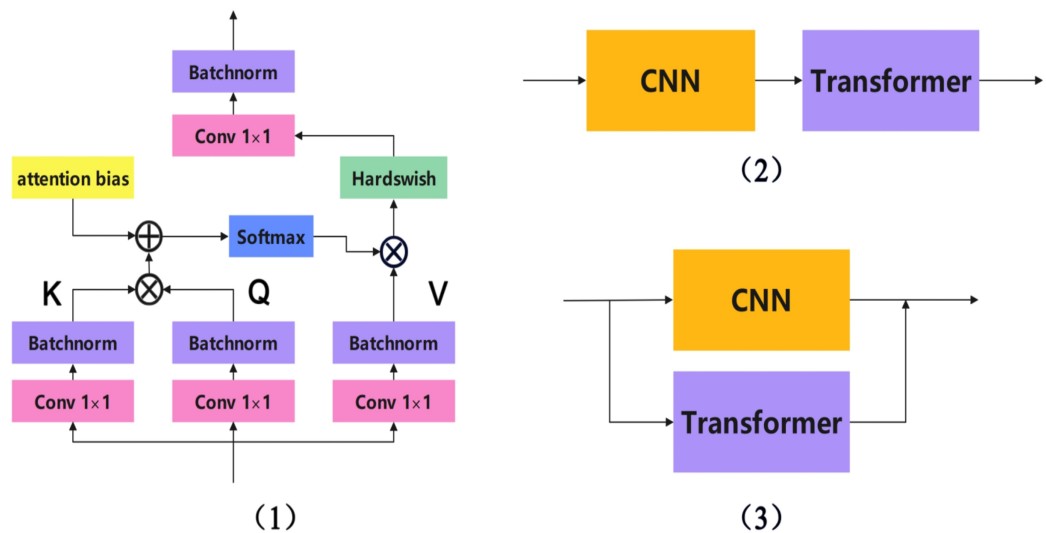

**Figure 2** **The three main CNN-transformer fusion methods.** (1) Early layer fusion, (2) sequential fusion, and (3) parallel fusion.               

examples of parallel fusion networks, including TransXNet (*Lou et al., 2023*), Mobile-Former (*Zhu et al., 2022*), Conformer (*Chen, Ningning & Zhaoxiang, 2021*), ScribFormer (*d'Ascoli et al., 2021*), LEFormer (*Chen et al., 2024*), Enriched (*Yoo et al., 2023*), and CTCNet (*Gao et al., 2023*). The three main CNN-Transformer fusion methods are shown in Fig. 2.

# PROPOSED METHOD

## Network structure

As introduced in "VIA Screening Procedure and Cervical Cancer Staging Criteria", the acetowhite images of low-grade cervical intraepithelial neoplasia (CIN1) only exhibit slight white changes, which may not be apparent. Current neural networks struggle to accurately and consistently capture these features. However, applying image enhancement (*An et al., 2022*; *Ren, Li & Xu, 2023*; *Tan et al., 2022*) to such images will undoubtedly deepen the color of the lesion area and increase the texture depth of the lesion. The color and texture of the lesion area are crucial indicators for distinguishing between CIN1 and CIN2. Without image enhancement, current neural networks find it challenging to accurately differentiate between normal samples and CIN1 samples. On the other hand, using image enhancement can affect the distinction between CIN1 and CIN2. To address this difficulty and maximize the accuracy of recognizing acetowhite samples of cervical intraepithelial neoplasia (CIN), this study designed the HMCFormer network and a dual-color space-based image enhancement technique. HMCFormer is a CNN+Transformer parallel fusion network. HMCFormer consists of two branches: the Transformer branch, the CNN branch. We input the original, non-enhanced sample images into the Transformer branch. The sample images enhanced using the dual-color space-based image enhancement

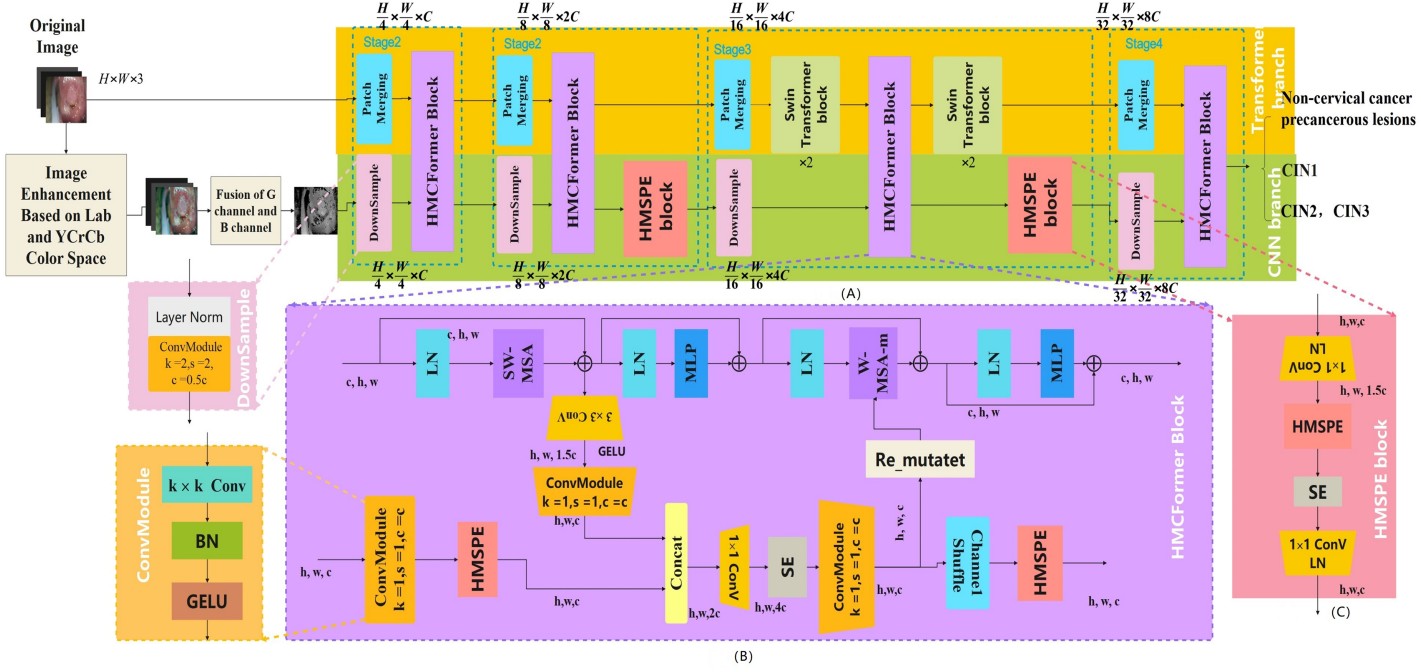

**Figure 3 HMCFormer network structure.** (A) HMCFormer; (B) HMCFormer block; (C) HMSPE block.

technique are converted into grayscale images, retaining only texture without color, and are then fed into the CNN branch, which excels at capturing detailed textures.

The aim of this study is to provide affordable cervical cancer screening for women in low- and middle-income regions. Thus, the designed network should be capable of running on low-cost computing devices. This article references networks such as Swin_Transformer_Tiny (*Liu et al., 2021*), EfficientNetV2_S (*Tan & Le, 2021*), and ConvNeXt_S (*Woo et al., 2023*). In the Transformer branch, the distribution of Transformer modules in each stage is (2, 2, 6, 2). In the CNN branch, the distribution of HMSPE modules in each stage is (1, 2, 2, 1). The model size is controlled to be around 12M. As shown in the Fig. 3.

## Dual color space-based image enhancement technology

The application of acetic acid solution to the cervical surface allows for the detection of cervical abnormalities based on changes in color and the degree of these changes (*Xie et al., 2021*). Low-grade CIN typically presents with faint and pale pinkish-white textures, which are difficult to observe with the naked eye and pose challenges for computer recognition. Most image enhancement techniques focus on modifying brightness, contrast, or enhancing high-frequency components to improve texture visibility. However, colposcopic images of the cervix are generally well-illuminated and high-resolution, making brightness modifications ineffective for texture enhancement. Increasing contrast may highlight the lesion areas, but it also significantly increases the roughness of the skin in these regions. Since skin

---

**Algorithm 1 Image enhancement.**

1. Input:

    Image: input image

2. Output:

    Img_E: output image

3. Functions:

    Lab_image <— Convert to Lab color space(image)

      L, A, B <— Split channels(Lab_image), split into three channels

    L_eq <— CLAHE(L)

    A_eq <— LT_CLAHE(A), apply Local Truncated Contrast Limited Adaptive Histogram Equalization to A channel

      B_eq <— LT_CLAHE(B), apply Local Truncated Contrast Limited Adaptive Histogram Equalization to B channel

    Lab_eq <— Merge channels(L_eq, A_eq, B_eq)

    YCrCb_image <— Convert to YCrCb color space(image)

    Y, Cr, Cb <— Split channels(YCrCb_image), split into three channels

    Y_eq <— CLAHE(Y)

    Cr_eq <— LT_CLAHE(Cr), apply Local Truncated Contrast Limited Adaptive Histogram Equalization to Cr channel

      Cb_eq <— LT_CLAHE(Cb), apply Local Truncated Contrast Limited Adaptive Histogram Equalization to Cb channel

    YCrCb_eq <— Merge channels(L_eq, A_eq, B_eq)

    Img_E <— YCrCb_eq×0.5 + Lab_eq×0.5

    Img_EG <— RGB to Gray

---

roughness is a critical factor in the grading of CIN, this approach complicates AI recognition. Furthermore, the similarity in color between the natural cervical texture and the lesion texture makes it difficult to enhance the lesion texture through high-frequency component enhancement.

To enhance the neural network's ability to recognize lesions more effectively, this article innovatively proposes increasing the color contrast between the lesion texture and the normal cervical skin texture. This approach highlights the lesion texture without significantly deepening it. We first decompose the image into the LAB and YCrCb color spaces. Using the locally truncated contrast-limited adaptive histogram equalization (LT_CLAHE) algorithm introduced in this study, we increase the contrast between the texture color and the background color. The LT_CLAHE algorithm achieves localized histogram equalization and is applied to the A, B, Cr, and Cb color channels, selectively equalizing certain colors while leaving others unchanged. This method effectively enhances the color difference between the lesion area and the normal cervical skin without significantly altering the overall image contrast, thereby avoiding the deepening of texture and increased roughness that could impair neural network recognition. Before use, we convert the enhanced image into a grayscale image, removing color information, and this grayscale image (Img_EG) is used as input for a CNN branch. This step eliminates color information, further reducing the adverse effects of image enhancement on AI-based grading of cervical intraepithelial neoplasia (Algorithm 1).

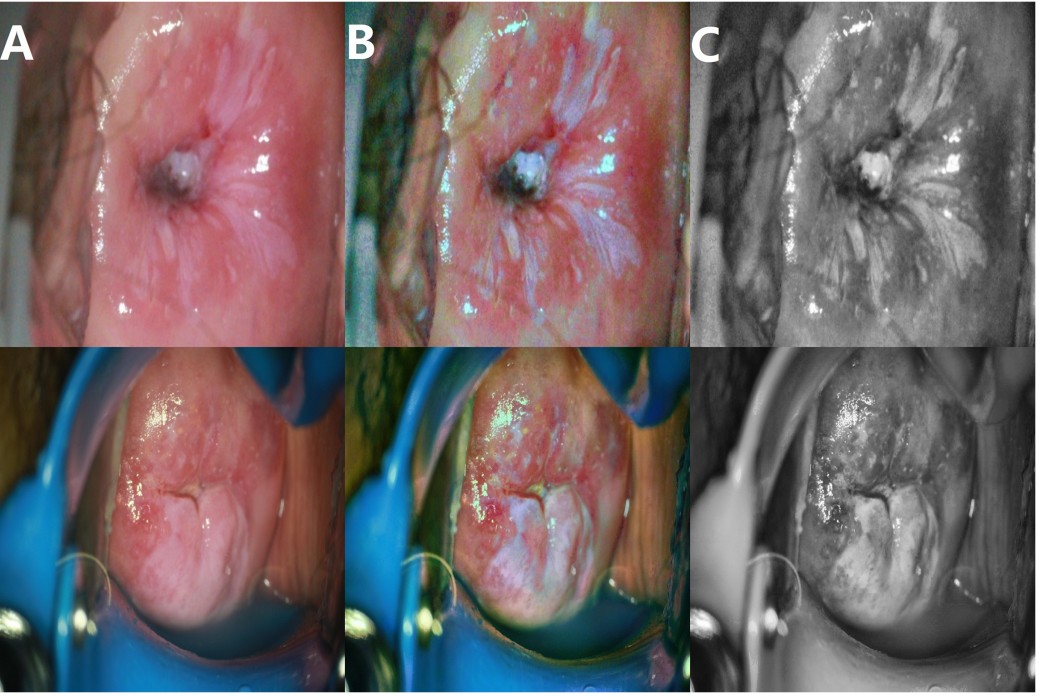

**Figure 4 Image enhancement effects.** (A) Original image; (B) enhanced image; (C) grayscale image.

The enhancement effect is shown in Fig. 4.

### Local truncated contrast-limited adaptive histogram equalization

Traditional contrast-limited adaptive histogram equalization (CLAHE) reallocates the pixel points of each sub-block by evenly distributing the number of clipped pixels to each gray level of the histogram.

The speStep 1: The input image was divided into non-overlapping sub-blocks of equal size, with M representing the number of pixels in each sub-block.

Step 2: Calculate the histogram. The histogram of the sub-blocks is represented by h(x), with x representing the gray level, which falls within the range of [0, L − 1], and L denoting possible gray levels.

Step 3: Calculate the clipLimit with the formula

$$cliplimit = \frac{M}{L} + \frac{(M - M/L)}{normClipLimit} \tag{1}$$

Step 4: Pixel point redistribution. For each sub-block, h(x) is clipped using the corresponding clipLimit value. Therefore, the clipped pixels are redistributed between 0 and D.

$$total\ E = \sum_{x=0}^{L-1} (max(h(x) - clipLimit, 0)) \tag{2}$$

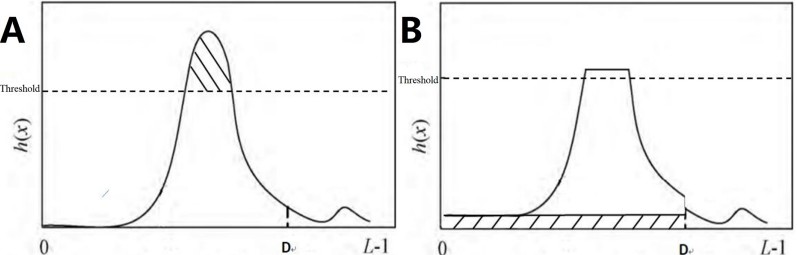

**Figure 5 LT CLAHE.** (A) Histogram before processing; (B) histogram after processing.

$$\text{avgBIncr} = \frac{total\ E}{D}, \quad D = \frac{2}{3} \times L \tag{3}$$

In the equation, total E refers to the total number of pixel values exceeding clipLimit. avgBIncr refers to the average number of pixels increased per gray level in the histogram. The above allocation process is repeated until all clipped pixels are redistributed, as shown in Fig. 5. If h′(x) denotes the histogram after pixel redistribution of h(x), then we have:

$$h'(x) = \begin{cases} h(x) & (x > D) \\ h(x) + avgBIncr & (x \leq D). \end{cases} \tag{4}$$

In this formula, upperLimit = clipLimit − avgBIncr.

Step 5: Histogram equalization. Histogram equalization was performed on h′(x), with f(x) expressing the equalization result.

Step 6: Reconstruction of pixel gray value. Based on f(x), the gray values of the central pixel points of each sub-block were obtained and used as a reference to calculate the gray values of each point in the output image by employing the bilinear interpolation technique.

## Downsampling module

In this article, different downsampling modules are employed in the Transformer branch and the CNN branch. In the Transformer branch, the original Patch Merging from Swin_T (*Liu et al., 2021*) is used. After passing through the Patch Merging layer, the height and width of the feature map are halved, while the depth is doubled. In the CNN branch, this article adopts the DownSample structure from ConvNext (*Woo et al., 2023*). A separate downsampling layer uses a 2 × 2 convolution with a stride of 2, and layer normalization (LN) is added before each downsampling layer to stabilize training. The downsampling steps are (4, 2, 2, 2).

## Transformer branch

The primary advantage of a parallel CNN+Transformers network lies in its ability to maximize the synergistic potential of both the Transformer and CNN branches, leveraging the strengths of both architectures while compensating for their respective weaknesses. However, significant differences exist between Transformers and CNNs in terms of feature

extraction and computational mechanisms, making it challenging to fully harness the collaborative advantages of both branches. When local features extracted by the CNN branch are fed into the Transformer branch, they may gradually be eliminated in the subsequent global feature extraction processes. This not only hinders the collaborative potential but can also lead to a decline in the performance of the Transformer branch.

To enhance the interaction between the CNN and Transformer branches, this article adopts the structure of the Swin Transformer (*Liu et al., 2021*) as the main framework for the Transformer branch.

The Swin Transformer employs a sliding window mechanism to perform self-attention calculations within local regions, a process that resembles the convolution operations in convolutional neural networks (CNNs). As a result, the features obtained from the Swin Transformer bear greater similarity to those extracted by CNNs compared to other Transformer architectures. The Swin Transformer features a pyramidal structure, which, like the similarly pyramidal CNN branch, captures features at comparable scales within each stage. This structural similarity facilitates the survival of features when integrated into the opposite branch during subsequent computations. Furthermore, the computational complexity of each layer in the Swin Transformer is reduced from the quadratic complexity of traditional Transformers to linear complexity. This improvement in computational efficiency is particularly advantageous when processing high-resolution medical images.

This article emulates the Swin Transformer by constructing the Transformer branch through repeated use of W-MSA-m, SW-MSA, and W-MSA modules. In stages 0, 1, and 3, an SW-MSA module is first utilized, followed by a W-MSA-m module. The formulation is as follows:

$$\hat{x}^l = SW\_MSA\big(LN\big(x^{l-1}\big)\big) + x^{l-1} \tag{5}$$

$$x^l = MLP\big(LN\big(\hat{x}^l\big)\big) + \hat{x}^l \tag{6}$$

$$\hat{x}^{l+1} = W\_MSA\_m\big(LN\big(x^l\big), LN\big(y^l\big)\big) + x^l \tag{7}$$

$$x^{l+1} = MLP\big(LN\big(\hat{x}^{l+1}\big)\big) + \hat{x}^{l+1}. \tag{8}$$

Here, SW-MSA refers to shifted window multi-head self-attention (*Liu et al., 2021*), and W-MSA represents window multi-head self-attention (*Liu et al., 2021*). W-MSA-m is an attention module improved upon W-MSA, specifically designed to effectively integrate information from the CNN branch. The details of this module will be thoroughly discussed in "HMCFormer Block". LN and MLP denote layer normalization and multi-layer perceptron, respectively.

In stage 2, we emulate the Swin Transformer Tiny by using six Transformer modules. At this stage, the features have become increasingly abstract, necessitating a greater number of Transformer modules to enhance the model's nonlinear representational capacity, enabling it to process more complex features and relationships. Since W-MSA-m is designed to receive information from the CNN branch, only one can be used per stage. Therefore, in stage 3, we add a Swin Transformer block both before and after the

HMCFormer block. Each Swin Transformer block consists of one W-MSA and one SW-MSA. The formulation for the Swin Transformer block is as follows:

$$\hat{x}^l = SW\_MSA\big(LN\big(x^{l-1}\big)\big) + x^{l-1} \tag{9}$$

$$x^l = MLP\big(LN\big(\hat{x}^l\big)\big)\big) + \hat{x}^l \tag{10}$$

$$\hat{x}^{l+1} = W\_MSA\big(LN\big(x^l\big), LN\big(y^l\big)\big) + x^l \tag{11}$$

$$x^{l+1} = MLP\big(LN\big(\hat{x}^{l+1}\big)\big)\big) + \hat{x}^{l+1}. \tag{12}$$

In our Transformer branch, the embedding dimension C is set to 32, the window size is set to 7, and the number of heads is configured as (3, 6, 12, 24).

## CNN branch

Multi-scale feature extraction enhances a model's ability to comprehensively understand images by detecting and describing target features across different scales, thereby improving the accuracy of tasks such as object detection and image classification (*Szegedy et al., 2024*; *Xu et al., 2024*). This capability is crucial for the recognition of cervical cancer images. Multi-level feature extraction has also gained popularity, as cross-stage feature transmission enables the extraction and reuse of features at various levels. This approach effectively captures a wide range of features, from low-level ones (*e.g.*, edges, textures) to high-level ones (*e.g.*, object shapes, semantic information). Such feature transmission and fusion strategies help improve gradient flow and alleviate the vanishing gradient problem, allowing deep networks to train more effectively and extract features more efficiently, as seen in the C3 structure of CSPNet (*Wang et al., 2020*) and the C2f structure in yolov8 (*Wang et al., 2024*). The thickness, texture, boundaries, and contours of lesion areas are critical features for determining the grade of cervical intraepithelial neoplasia, making the ability to comprehensively extract multi-scale and multi-level features an urgent necessity.

To achieve comprehensive multi-scale and multi-level feature extraction with minimal computational overhead, this article introduces the innovative HMSPE module. As illustrated in the figure, HMSPE effectively integrates multi-scale and multi-level feature extraction. Given an input feature vector, it is first passed through a 1 × 1 convolution and then divided into four sub-channels: x1, x2, x3, and x4 each having the same feature size. To accomplish multi-scale feature extraction, we employ a combination of 5 × 5 convolution, 3 × 3 convolution, and 3 × 3 dilated convolution. Unlike other multi-scale feature extraction approaches, we draw inspiration from the MobileNetV2 (*Sandler et al., 2018*) block structure by incorporating 1 × 1 convolutions for channel expansion and compression, achieving efficient feature representation and computation. While this operation may seem to increase computational complexity, our experiments show that the HMSPE module exhibits strong expressive power, requiring only one or two HMSPE modules per stage to achieve satisfactory performance. This is in contrast to prior work, which often necessitates multiple applications of multi-scale feature extraction modules within each stage to achieve comparable expressive power. Overall, this approach significantly reduces computational cost.

Let the computation of the $5 \times 5$ Conv block be denoted as (x), the computation of the $3 \times 3$ Conv block as, and the computation of the $3 \times 3$ Dilated Conv block as. The input feature vector is denoted as x.

$$f_{5\times5}(x) = \text{Conv}_{1\times1}^{0.5C\to0.25C}\big(\text{GELU}(\text{Conv}_{5\times5}^{0.25C\to0.5C}(x))\big) + x \tag{13}$$

$$f_{3\times3}(x) = \text{Conv}_{1\times1}^{0.5C\to0.25C}\big(\text{GELU}(\text{Conv}_{3\times3}^{0.25C\to0.5C}(x))\big) + x \tag{14}$$

$$f_{3\times3}^d(x) = \text{Conv}_{1\times1}^{0.5C\to0.25C}\Big(\text{GELU}\big(\text{DilatedConv}_{3\times3,d=1}^{0.25C\to0.5C}(x)\big)\Big) + x. \tag{15}$$

For the x1 sub-channel, no operation is performed. The x2 sub-channel utilizes a $5 \times 5$ Conv block, while the x3 sub-channel follows the sequence of a $3 \times 3$ ConV block → $3 \times 3$ DilatedConV block → $3 \times 3$ DilatedConV block. The x4 sub-channel applies a sequence of $3 \times 3$ ConV block → $3 \times 3$ ConV block → $3 \times 3$ DilatedConV block. When the dilation factor d = 1, the receptive field of a $3 \times 3$ Dilated Conv is similar to that of a $5 \times 5$ Conv. Thus, two $3 \times 3$ Dilated Conv blocks with d = 1 are used as a substitute for the $5 \times 5$ Conv. Drawing inspiration from the progressive convolution and feature fusion concepts of the Res2Net block (*Gao et al., 2019*), further scale expansion is achieved. The formulation of the HMSPE module is as follows:

$$y1 = \text{PEM}(\text{Concat}(x1, f_{5\times5}(x2))) \tag{16}$$

$$y2 = \text{PEM}\big(\text{Concat}\big(f_{3\times3}(x3), f_{3\times3}^d(f_{5\times5}(x2) + f_{3\times3}(x3))\big)\big) \tag{17}$$

$$y3 = \text{PEM}\big(\text{Concat}\big(f_{3\times3}(f_{3\times3}(x4)), f_{3\times3}^d(f_{3\times3}^d(f_{5\times5}(x2) + f_{3\times3}(x3)))\big)\big) \tag{18}$$

$$y4 = \text{PEM}\big(\text{Concat}\big(f_{3\times3}(x4), f_{3\times3}^d(f_{3\times3}(f_{3\times3}(x4)) + f_{3\times3}^d(f_{5\times5}(x2) + f_{3\times3}(x3)))\big)\big) \tag{19}$$

$$Y = \text{Cancat}(y1 + y2, y3 + y4) + x.$$

The pixel-wise excitation module (PEM), as illustrated in the Fig. 6, is designed to independently excite and enhance each pixel of the input feature map. After concatenating the multi-scale and multi-level feature vectors, they are fed into the PEM module, which enhances the representational capacity of high-dimensional features, aiding the model in better capturing complex image information. LN denotes layer normalization, and Y represents the output of the HMSPE module.

The structure in which the CNN branch receives inputs from the Transformer branch will be detailed in "HMCFormer Block". CNN networks require a large number of channels to capture the rich features and texture information present in images. Typically, CNNs need more channels than transformers. However, to facilitate seamless cross-fusion, it is important to ensure that the number of channels in the Transformer branch and the CNN branch are as consistent as possible. To increase the number of channels in the CNN branch, an additional HMSPE module is introduced in both Stage 2 and Stage 3, as shown in Fig. 3C. The HMSPE module, inspired by the MobileNetV3 (*Koonce & Koonce, 2021*) architecture, first doubles the number of channels using a $1 \times 1$ convolution, followed by an HMSPE module, and then an SE module with another $1 \times 1$ convolution to align the number of channels with the Transformer branch. This approach not only ensures channel consistency between the CNN and Transformer branches but also alleviates the demand for more feature channels in the CNN network.

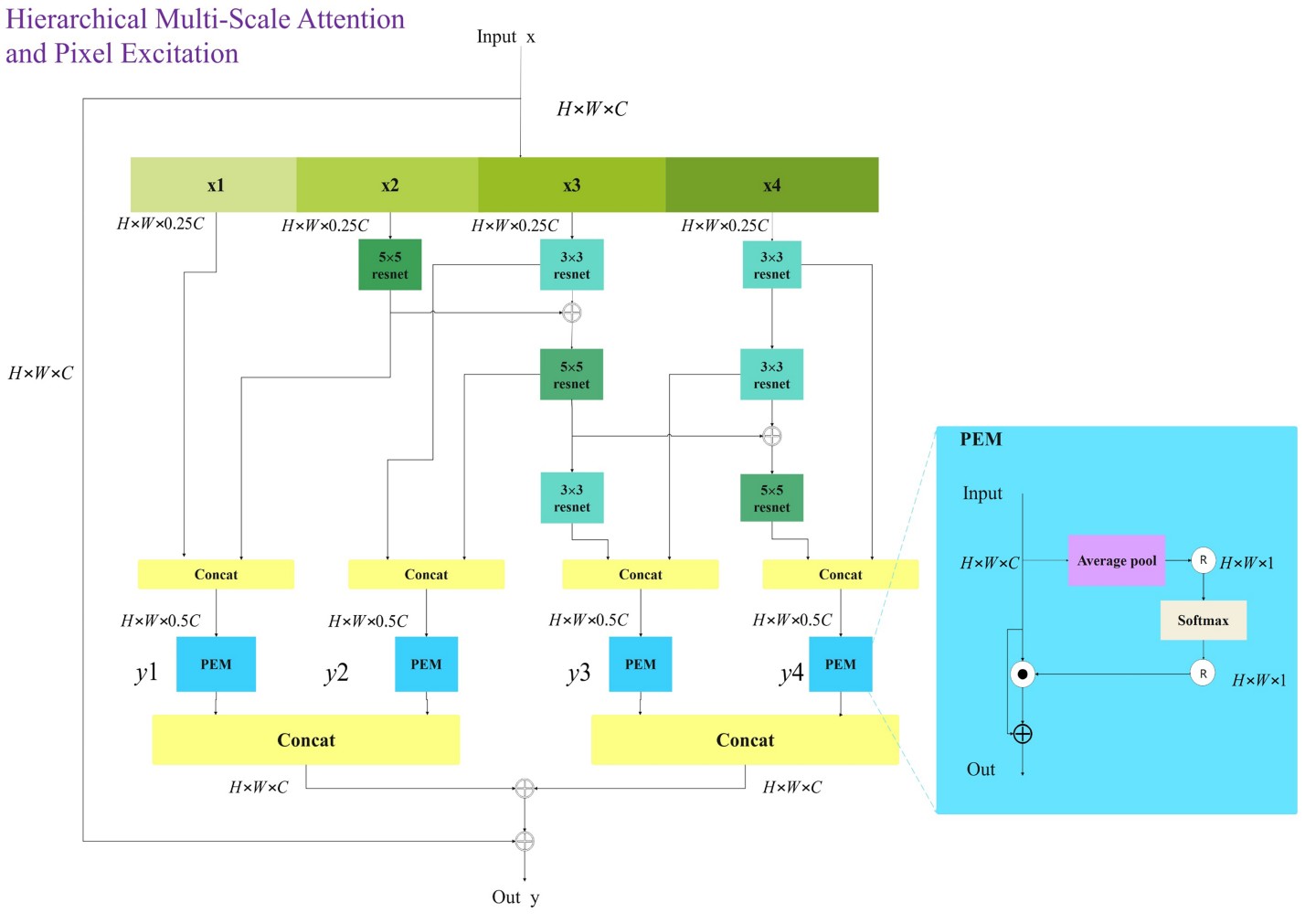

**Figure 6 Hierarchical multi-scale pixel excitation (HMSPE) module.**

In our CNN branch, the size of the embedded spatial dimension C is set to 32.

## HMCFormer block

The HMCFormer block is a crucial module for enabling the exchange of information between the Transformer and CNN branches. As discussed in *d'Ascoli et al. (2021)*, *Chen et al. (2024)*, *Yoo et al. (2023)*, *Gao et al. (2023)*, and *Xu et al. (2024)*, integrating features from Transformers and CNNs can leverage their synergistic capabilities. Different fusion methods can have a significant impact on the overall network performance. However, no existing work has proposed a scientific approach to evaluate whether a fusion method effectively harnesses the synergistic potential of both architectures. To address this, we propose using mutual information, a concept from information theory, to assess the effectiveness of different fusion strategies.

Let X denote the features from the Transformer branch and Y denote the features from the CNN branch, with the mutual information between X and Y denoted as I(X;Y). The mutual information is defined as:

$$I(X;Y) = H(X) + H(Y) - H(X, Y) \tag{20}$$

where H(X) is the entropy of the random variable X representing the information content of the Transformer branch, H(Y) is the entropy of the random variable Y, representing the information content of the CNN branch, and H(X, Y) is the joint entropy of the random variables X and Y.

A comprehensive analysis of I(X;Y), H(X), and H(Y) can effectively evaluate whether the network fully leverages the synergistic potential of the two branches. Ideally, when both H(X) and H(Y) are relatively large, I(X;Y) should satisfy the condition $I(X;Y) \leq H(X) + H(Y) - Max(H(X), H(Y))$. High values of H(X) and H(Y) indicate a high degree of feature diversity in the Transformer and CNN branches, which enhances the model's ability to understand different aspects of the input data, making it advantageous for classification tasks. If I(X;Y) is slightly less than $H(X) + H(Y) - Max(H(X), H(Y))$, this suggests a strong correlation between the Transformer and CNN features, with minimal redundancy, indicating that the network is effectively utilizing the synergistic potential of both branches. Conversely, if $I(X; Y) > H(X) + H(Y) - Max(H(X), H(Y))$, it indicates a high degree of redundancy between X and Y, leading to inefficient network performance. On the other hand, if $I(X; Y) < H(X) + H(Y) - max(H(X), H(Y))$, it suggests that there is little correlation between X and Y, indicating that the network has not fully exploited the synergistic capabilities of the two branches.

As demonstrated in our experiments, which will be detailed in "Comparative Performance of HMCFormer", traditional element-wise addition or concatenation operations do not yield optimal results. Previous studies (*d'Ascoli et al., 2021*; *Yoo et al., 2023*; *Xu et al., 2024*) have proposed related weighted fusion methods. However, these methods tend to diminish the influence of the CNN branch on the Transformer branch, resulting in a lower H(X) value. In Transformer networks, the distinctiveness of the features extracted from image patches tends to diminish as the network depth increases, causing them to become increasingly similar or indistinguishable (*Diko et al., 2024*). This characteristic necessitates a larger sample size for Transformer networks compared to CNNs. Integrating information from the CNN branch into the Transformer branch can effectively mitigate this issue. The related weighted fusion method (*d'Ascoli et al., 2021*; *Yoo et al., 2023*; *Xu et al., 2024*) involves first fusing the Transformer and CNN branches using various algorithms to obtain a fused feature map, and then adding this fused feature map to both the Transformer and CNN branches. Our research found that this approach is highly inefficient; the positive impact on the Transformer branch primarily comes from the information in the fused feature map originating from the CNN branch. The mechanism by which this fusion method works is as follows: during the fusion process between the Transformer and CNN branches, a significant amount of information from the CNN branch is lost. When the fused feature map is subsequently added to the

Transformer branch, the information from the Transformer branch holds a relative advantage over the CNN-derived information in the fused feature map. This ensures that the Transformer can largely maintain its informational independence while still acquiring the valuable insights from the CNN branch, preventing the Transformer network from losing its uniqueness as network depth increases. Directly fusing the information from the CNN branch, which is at an informational disadvantage, with the Transformer branch is a more efficient approach compared to the related weighted fusion method.

To maximize the synergistic potential of both branches, this article introduces the concepts of adaptive preprocessing and superiority-inferiority fusion. Adaptive Preprocessing: Given the distinct differences in features and computational mechanisms between Transformers and CNNs, information exchange between the CNN branch and the Transformer branch should first pass through an adaptive structure before fusion is implemented. Superiority-inferiority fusion: The information from the CNN branch that is fed into the Transformer branch should be at a moderate disadvantage relative to the information from the Transformer branch, and vice versa. This approach ensures the independence of each branch's information, maintaining relatively high values for H(X) and H(Y). The fusion method proposed in _Yoo et al. (2023)_ aligns with the concepts of adaptive preprocessing and disadvantageous fusion introduced in this article. However, this method remains somewhat rudimentary and offers considerable room for improvement.

In the proposed model, the fusion of CNN branch information into the Transformer branch is not performed at the end of each stage, as is common in other parallel networks, but instead occurs within the second attention module of each HMCFormer Block. The global feature extraction capability of ViT-based networks significantly improves with increased network depth. By conducting the fusion in the second attention module, the information from the CNN branch undergoes only half the attention computation compared to the original Transformer information in each stage, thus achieving the intended superiority-inferiority fusion. Compared to the fusion method in _d'Ascoli et al. (2021)_, _Yoo et al. (2023)_, and _Xu et al. (2024)_, the approach proposed in this article ensures that the Transformer branch retains its informational advantage while fully incorporating the information from the CNN branch. The features from the CNN branch undergo adaptive preprocessing through a layer normalization (LN) and multi-layer perceptron (MLP) module before being fed into the second attention module (W_MSA_m) of the HMCFormer block. The W_MSA_m module represents an enhancement of the original W_MSA, enabling it to effectively integrate information from the CNN branch.

We divide the feature map into multiple windows. Let X represent the feature map from the Transformer branch and y represent the feature map from the CNN branch. The expression for the module is as follows:

$$Y = \mathrm{LN}(\mathrm{MLP}(y))$$

$$Q_i^h = W_Q^h \cdot X_i;\ K_i^h = W_K^h \cdot X_i;\ V_i^h = W_V^h \cdot X_i \tag{21}$$

$$q_i^h = W_Q^h \cdot Y_i;\ k_i^h = W_K^h \cdot Y_i;\ v_i^h = W_V^h \cdot Y_i \tag{22}$$

$$\text{Attention}_i^h = \text{softmax}\left(\frac{Q_i^h\left(K_i^h\right)^T + q_i^h\left(k_i^h\right)^T}{\sqrt{d_k}}\right)\left(V_i^h + v_i^h\right). \tag{23}$$

Let $X_i$ denote the input features of the feature map $X$ within window $i$, and $Y_i$ denote the input features of the feature map $Y$ within window $i$. $W_Q$, $W_K$, $W_V$ are the weight matrices for the linear transformations.

Compared to traditional element-wise addition or concatenation operations, using the proposed $W\_MSA\_m$ module for fusion enables more fine-grained feature integration, allowing for a more detailed merging of the features from the two tensors. This approach helps capture more complex feature relationships. According to the evaluation metrics proposed in this article, the H(X) value obtained using this fusion method is significantly higher than that of other fusion methods.

When integrating the information from the Transformer branch into the CNN branch, we replicated the aforementioned approach by performing the fusion between the first HMSPE module and the second HMSPE module within each HMCFormer Block. Transformers and CNNs exhibit significant differences in feature representation and computational mechanisms. To accommodate these differences, we processed the feature tensor from the Transformer branch using a 3 × 3 Conv block followed by a 1 × 1 Conv block for adaptive preprocessing before concatenating it with the output feature tensor of the HMSPE module. In each stage, the information from the Transformer branch undergoes nearly half the HMSPE computations compared to the original CNN branch, leading to the Transformer-derived information being at a disadvantage post-concatenation. This step effectively achieves the goal of integrating the Transformer branch's information at a disadvantageous position. The fused feature tensors from the two branches are then integrated through a MobileNetV2 Block structure.

Let $Y_h$ denote the feature tensor output from the HMSPE module, and X denote the feature tensor from the Transformer branch.

$$F(X) = \text{Conv}_{1\times1}^{1.5C \rightarrow C}\left(\text{GELU}\left(\left(\text{Conv}_{3\times3}^{C \rightarrow 1.5C}(X)\right)\right)\right) \tag{24}$$

$$F1 = \text{Concat}(Y_h, F(X)) \tag{25}$$

$$Y_{out} = \text{Shuffle}\left(\text{Conv}_{1\times1}^{4C \rightarrow C}\left(\text{SE}\left(\text{Conv}_{1\times1}^{2C \rightarrow 4C}(F1)\right)\right)\right). \tag{26}$$

Let $Y_{out}$ denote the output after fusion in the CNN branch, and SE represent the SE attention module. According to the evaluation metrics proposed in this article, the H(Y) value obtained using this fusion method is slightly higher than that of the fusion method described in *d'Ascoli et al. (2021)* and *Yoo et al. (2023)*.

Using the fusion method proposed in this article, the three metrics—I(X;Y), H(X), and H(Y)—all show improved performance. This indicates that the proposed fusion approach effectively leverages the synergistic advantages of both the CNN and Transformer networks.

**Table 1 Data sources.**

| Name | Database | Identifier | URL |
|---|---|---|---|
| Google cervical cancer segmentation | GitHub | google/cervical-cancer-segmentation | https://www.kaggle.com/datasets/madhurieseth/cervical-cancer-my-dataset |
| Intel & MobileODT cervical cancer screening dataset | Kaggle | intel-mobileodt-cervical-cancer-screening | https://www.kaggle.com/competitions/intel-mobileodt-cervical-cancer-screening/data |
| NIH cervical cancer dataset | The cancer imaging archive (TCIA) | TCIA.2016.0J5042YQ | https://www.cancerimagingarchive.net Official Website: https://www.cancerimagingarchive.net/ |

## EXPERIMENTS

### Dataset introduction

The proposed HMCFormer was trained and tested on publicly available cervical image datasets (https://www.kaggle.com/datasets/madhurieseth/cervical-cancer-my-dataset; https://www.kaggle.com/competitions/intel-mobileodt-cervical-cancer-screening/data; https://www.cancerimagingarchive.net). As shown in Table 1.

After organizing and classifying these three public datasets, 5,165 samples suitable for VIA screening were obtained, which we refer to as the PCC5000 dataset. This dataset includes 2,066 samples of non-cervical intraepithelial neoplasia, 1,212 samples of CIN1, 1,225 samples of CIN2, and 662 samples of CIN3. All images were re-annotated by professional gynecologists.

### Data preprocessing

To evaluate the model performance more comprehensively, we applied a five-fold cross-validation method on 5,165 samples. Specifically, the dataset was divided into five subsets of equal size. In each iteration, one subset was chosen as the validation set, another as the test set, and the remaining three subsets were combined as the training set. This process was repeated 5 times, ensuring that each subset was used as the validation and test set once. Finally, the results from all five iterations were averaged to obtain the overall performance metrics of the model.

During the data preprocessing stage, we applied the following specific methods and steps to augment the input images: (1) Resized the images to $260 \times 260$; (2) Randomly cropped the images to $224 \times 224$; (3) Randomly rotated the images within a range of $[-15, 15]$ degrees; (4) Randomly flipped the images with a 50% probability; (5) Converted the images to tensors and normalized them.

### Implementation details

The experiments were conducted on an Intel ® Core ™ i7-7700K with a CPU of 4.2 GHz and a RTX2080Ti graphics card with 11 GB of memory. The neural network experiments were carried out using the PyTorch 1.9.0 and Ubuntu18.0 LTS software environment. The network input was set to $3 \times 224 \times 224$, with a batch size of 24, and Epochs = 300. Learning Rate Scheduling.

Initially, the model was pre-trained using the ImageNet1K dataset for 50 epochs. Subsequently, we utilized the weights obtained from the pre-trained model on ImageNet1K for transfer learning and conducted further training on our dataset.

## Evaluation metrics

To verify the effectiveness of the proposed network, we computed evaluation criteria such as accuracy, specificity, sensitivity, precision, recall, and F1-score. In these metrics, TP refers to true positive, TN to true negative, FP to false positive, and FN to false negative.

$$\text{Accuracy} = \frac{TP + TN}{TP + FP + FN + TN} \tag{27}$$

$$\text{Sensitivity} = \frac{TP}{TP + FN} \tag{28}$$

$$\text{Specificity} = \frac{TN}{TN + FP} \tag{29}$$

$$\text{F1-score} = 2 \times \frac{\text{Recall} \times \text{Precision}}{\text{Recall} + \text{Precision}} \tag{30}$$

$$\text{F1-score} = 2 \times \frac{\text{Recall} \times \text{Precision}}{\text{Recall} + \text{Precision}} \tag{31}$$

$$\text{Kappa} = \frac{P_o - P_e}{1 - P_e} \tag{32}$$

- $P_o = \dfrac{TP + TN}{Total\ Samples}$

- $P_e = \dfrac{(TP + FP) \cdot (TP + FN) + (FN + TN) \cdot (FP + TN)}{(Total\ Samples)^2}$

$$\text{AUC} = \frac{\sum_{i=1}^{N^+} \sum_{j=1}^{N} I(p_i > p_j)}{N_+ \cdot N_-}. \tag{33}$$

- $N_+$: Number of positive samples, $N_-$: Number of negative samples

- $p_i$: Predicted probability of the i-th positive sample, $p_j$: Predicted probability of the $j$-th negative sample.

## Comparative performance of HMCFormer

To compare the network performance of HMCFormer, we selected 23 representative neural networks: ResNet50 (*Allmendinger et al., 2022*), EfficientNet (2021) (*Tan & Le, 2021*), ConvNeXt (2022) (*Woo et al., 2023*), ConvNeXtv2 (2023) (*Woo et al., 2023*), MobileNetV3 (2021) (*Koonce & Koonce, 2021*), Swin_T (2021) (*Liu et al., 2021*), SwinV2_T (2022) (*Liu et al., 2022*), Pvt_v2 (2022) (*Wang et al., 2022*), FastViT (2023) (*Vasu et al., 2023*), EfficientFormer (2022) (*Li et al., 2022*), ConTNet (2023) (*Liu et al., 2023a*), AgentSwin (2024) (*Han et al., 2023*), AgentPVT (2024) (*Han et al., 2023*), Mlla (2024) (*Han et al., 2024*), CASViT (2024) (*Zhang et al., 2024*), SwiftFormer (2023) (*Shaker et al., 2023*), PoolFormer (2023) (*Yu et al., 2022*), EfficientViT (2023) (*Liu et al., 2023b*), Vmanba (2024) (*Liu et al., 2024*), WLAMFormer (2025) (*Feng et al., 2025*), Conformer (2021) (*Chen, Ningning & Zhaoxiang, 2021*), and TransXNet (2024) (*Lou et al., 2023*). Among these, there are five CNN-based networks, 11 Transformer-based networks, and

**Table 2 Comparison of binary classification performance.**

| Network model | Params (M) | FLOPs (G) | Accuracy | Sensitivity | Specificity | Error rate | Precision | F1 |
|---|---|---|---|---|---|---|---|---|
| Resnet50×0.75 | 16.8 | 3.14 | 0.806 | 0.841 | 0.790 | 0.193 | 0.661 | 0.741 |
| EfficientNet-b3 | 12.3 | 2.01 | 0.866 | 0.702 | 0.947 | 0.133 | 0.866 | 0.775 |
| ConvNeXt-small | 12.4 | 2.13 | 0.930 | 0.859 | 0.962 | 0.071 | 0.918 | 0.895 |
| ConvNeXtv2-Tiny | 15.3 | 2.32 | 0.916 | 0.837 | 0.954 | 0.083 | 0.899 | 0.867 |
| MobileNetV3-L | 4.2 | 0.28 | 0.813 | 0.682 | 0.876 | 0.187 | 0.728 | 0.704 |
| Swin_Tiny×0.5 | 13.4 | 2.18 | 0.814 | 0.803 | 0.819 | 0.185 | 0.684 | 0.739 |
| SwinV2_T×0.5 | 12.2 | 2.03 | 0.860 | 0.745 | 0.916 | 0.139 | 0.813 | 0.777 |
| Pvt_v2_b1 | 12.8 | 2.04 | 0.897 | 0.859 | 0.934 | 0.102 | 0.926 | 0.892 |
| SwiftFormer-L1 | 12.05 | 1.604 | 0.928 | 0.850 | 0.966 | 0.071 | 0.924 | 0.886 |
| EfficientViT-M5 | 12.47 | 0.525 | 0.918 | 0.898 | 0.938 | 0.081 | 0.933 | 0.915 |
| PoolFormer-S12 | 11.9 | 1.813 | 0.922 | 0.898 | 0.941 | 0.0076 | 0.941 | 0.920 |
| CAS-ViT-m | 12.03 | 1.66 | 0.920 | 0.888 | 0.948 | 0.081 | 0.933 | 0.915 |
| Agent-Swin-t×0.75 | 15.5 | 2.46 | 0.923 | 0.906 | 0.935 | 0.076 | 0.912 | 0.909 |
| Agent-PVT-T | 11.6 | 2.0 | 0.911 | 0.878 | 0.938 | 0.092 | 0.933 | 0.908 |
| FastVit-SA12 | 10.9 | 1.92 | 0.922 | 0.898 | 0.958 | 0.077 | 0.969 | 0.932 |
| EfficientFormer-S | 13.6 | 2.03 | 0.933 | 0.872 | 0.963 | 0.066 | 0.920 | 0.895 |
| ConTNet-XS | 12.2 | 1.94 | 0.930 | 0.861 | 0.964 | 0.069 | 0.921 | 0.890 |
| Conformer×0.5 | 18.1 | 2.88 | 0.920 | 0.863 | 0.948 | 0.079 | 0.891 | 0.877 |
| TransXNet-L | 12.8 | 1.83 | 0.947 | 0.913 | 0.963 | 0.052 | 0.924 | 0.919 |
| Vmanba-T×0.5 | 15.1 | 2.84 | 0.933 | 0.872 | 0.963 | 0.066 | 0.920 | 0.895 |
| WLAM**Former-L** | **13.5** | **2.847** | 0.943 | 0.918 | 0.970 | 0.062 | 0.921 | 0.919 |
| MLLA-t×0.75 | 13.5 | 2.68 | 0.937 | 0.978 | 0.907 | 0.062 | 0.887 | 0.930 |
| **HMCFormer (Ours)** | **12.5** | **1.78** | **0.974** | **0.981** | **0.968** | **0.025** | **0.958** | **0.970** |

Note:
The bold entries in the comparison table indicate the results of the model proposed in this article.

seven hybrid networks. To correspond to the scale of the proposed network in this article, we chose models around 12M in size for all 16 neural networks.

As shown in Table 2, HMCFormer significantly outperforms other neural network architectures across various metrics in binary classification. Since the primary goal of this article is to achieve intelligent cervical cancer screening, accurately identifying all potential cervical cancer cases is crucial, particularly in terms of sensitivity and accuracy. HMCFormer achieves an accuracy of 97.4%, sensitivity of 98.1%, and an F1-score of 97.0%, demonstrating superior performance across key metrics. Compared to other top-performing Transformer models, EfficientFormer-S lags behind HMCFormer by 3.1% in accuracy, 10.9% in sensitivity, and 7.5% in F1-score. Additionally, HMCFormer excels in reducing computational resources, with 1.1M fewer parameters and 0.25G fewer FLOPs than EfficientFormer-S. For CNN architectures, ConvNeXt-small is the best-performing model, but HMCFormer surpasses ConvNeXt-small in accuracy, sensitivity, and F1-score by 4.4%, 12.2%, and 8.1%, respectively. Although HMCFormer has 1M more parameters than ConvNeXt-small, it still reduces FLOPs by 0.35G, showcasing its high computational efficiency. In the case of CNN+Transformer fusion models, TransXNet-L is the

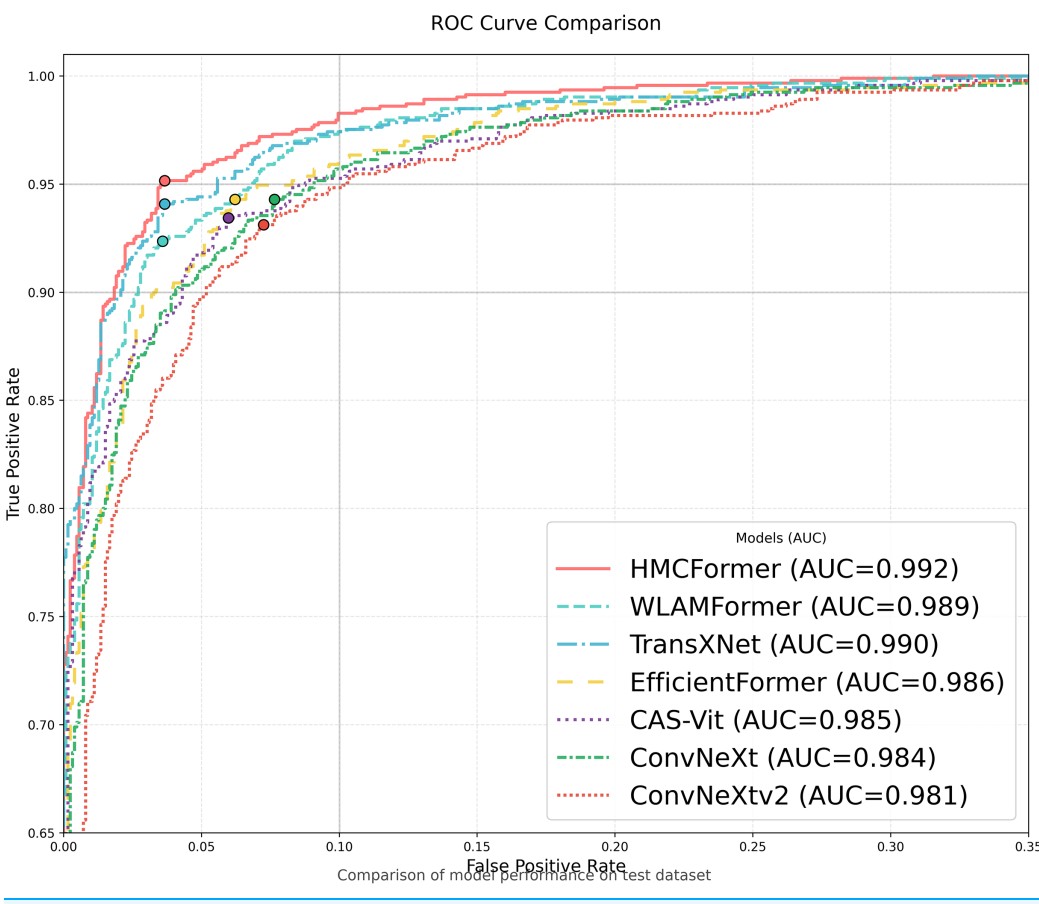

**Figure 7** Model ROC comparison.   

best-performing model other than HMCFormer. However, HMCFormer outperforms TransXNet-L in accuracy, sensitivity, and F1-score by 2.7%, 6.8%, and 5.1%, respectively. Moreover, HMCFormer demonstrates significant advantages in both parameters and FLOPs, with 1M fewer parameters and 0.95G fewer FLOPs than TransXNet-L. Figure 7 presents a comparison of ROC curves for several models, while Fig. 8 shows the accuracy of each model in binary classification on the PCC5000 dataset, with the size of the bubbles representing the number of parameters (Params) for each model.

Since patients in the CIN1 stage are likely to recover naturally, while those in the CIN2 and CIN3 stages are at a high risk of progressing to cancer, we often categorize CIN into low-grade (CIN1) and high-grade (CIN2/3). Given that other networks perform poorly in binary classification for cervical cancer detection, we only compare the networks that achieve an accuracy of over 92% in binary classification for the three-class classification (no precancerous lesion, CIN1, CIN2/3). Since each class is critically important, we choose macro-averaged metrics as the evaluation criteria.

From the analysis of Table 3, it is clear that our proposed HMCFormer network continues to exhibit superior performance in the three-class classification task. ConvNeXt_small is the best-performing CNN model; however, the model proposed in this

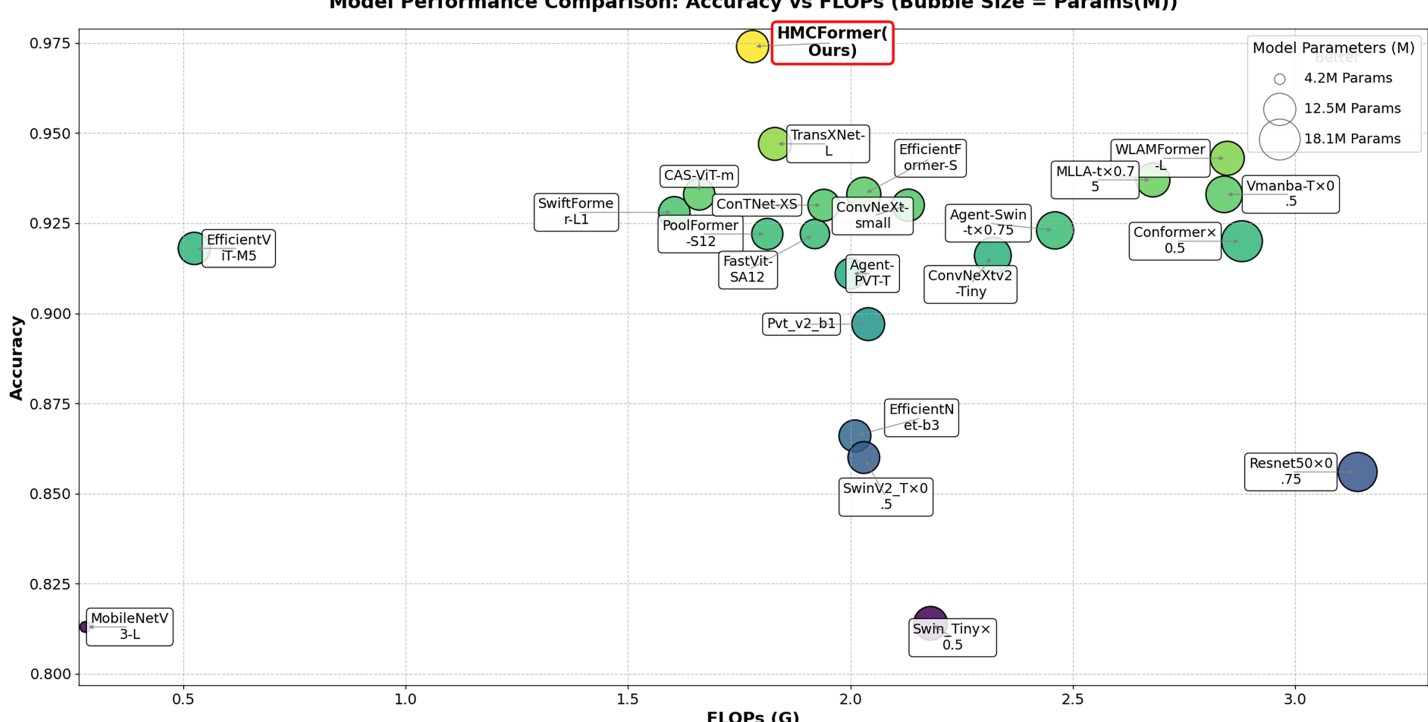

**Figure 8** Comparison of binary classification accuracy among different models, with bubble size representing the size of params.

**Table 3 Comparison of ternary classification performance.**

| Network model | Params (M) | FLOPs (G) | Type | Accuracy | Sensitivity | Precision | F1 | Kappa |
|---|---|---|---|---|---|---|---|---|
| ConvNeXt_small | 12.4 | 2.13 | CNN | 0.912 | 0.877 | 0.878 | 0.878 | 0.817 |
| SwiftFormer-L1 | 12.05 | 1.604 | Transformer | 0.881 | 0.819 | 0.854 | 0.834 | 0.789 |
| PoolFormer-S12 | 11.9 | 1.813 | | 0.919 | 0.873 | 0.901 | 0.884 | 0.859 |
| FastVit-SA12 | 10.9 | 1.92 | | 0.908 | 0.897 | 0.905 | 0.899 | 0.845 |
| Agent-Swin-t×0.75 | 15.5 | 2.46 | | 0.918 | 0.879 | 0.899 | 0.887 | 0.845 |
| EfficientFormer-S | 13.6 | 2.03 | | 0.921 | 0.888 | 0.886 | 0.887 | 0.864 |
| TransXNet-L | 12.8 | 1.83 | Hybrid | 0.931 | 0.903 | 0.904 | 0.903 | 0.880 |
| Vmanba-T×0.5 | 15.1 | 2.84 | | 0.928 | 0.891 | 0.904 | 0.897 | 0.875 |
| WLAMFormer-L | 13.5 | 2.847 | | 0.931 | 0.890 | 0.910 | 0.899 | 0.879 |
| MLLA-t×0.75 | 13.5 | 2.68 | | 0.927 | 0.896 | 0.900 | 0.898 | 0.871 |
| Conformer×0.5 | 18.1 | 2.88 | | 0.914 | 0.889 | 0.857 | 0.870 | 0.821 |
| **HMCFormer (Ours)** | **12.5** | **1.78** | | **0.948** | **0.928** | **0.919** | **0.923** | **0.898** |

**Note:**
The bold entries in the comparison table indicate the results of the model proposed in this article.

article surpasses it by 3.6% in accuracy, 5.1% in sensitivity, and 4.5% in F1-score, while maintaining similar parameter count and reducing FLOPs by 1.83G. EfficientFormer-S is the best-performing Transformer model, but our proposed model exceeds it by 2.7% in accuracy, 4.0% in sensitivity, and 3.6% in F1-score, with 1.1 million fewer parameters and a

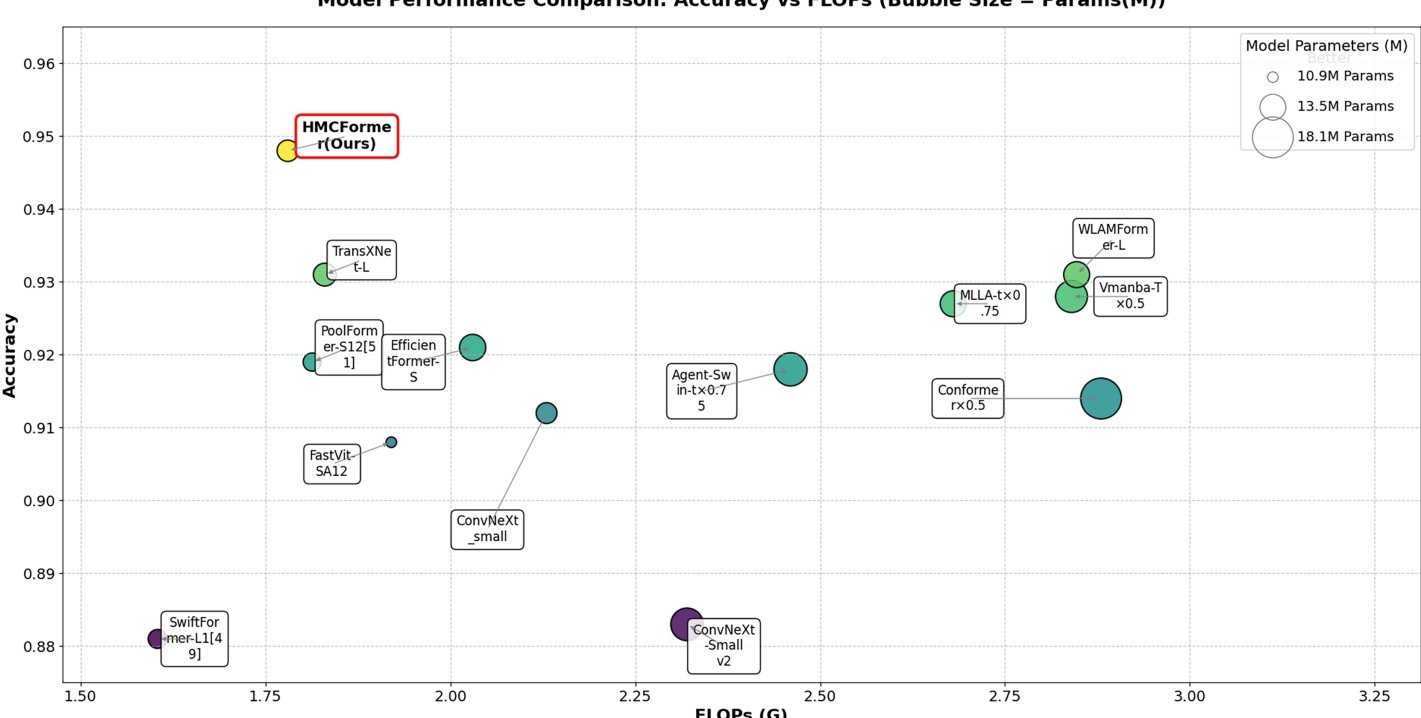

**Figure 9 Comparison of tri-class classification accuracy among different models, with bubble size representing the size of params.**

reduction of 0.25G in FLOPs. TransXNet-L is the best-performing CNN+Transformer hybrid network aside from the network proposed in this article; however, our model surpasses it by 1.7% in accuracy, 2.5% in sensitivity, and 2.0% in F1-score, while reducing parameters by 0.3 million and decreasing FLOPs by 0.95G. The results in the table further highlight the efficiency of HMCFormer, which achieves the highest accuracy (0.948) while maintaining a relatively low number of parameters (12.5M) and FLOPs (1.78G). This demonstrates that HMCFormer excels in not only accuracy but also computational efficiency compared to its counterparts. Figure 9 displays the accuracy of each model in multi-class classification (three classes) on the PCC5000 dataset, where the size of the bubbles represents the number of parameters (params) for each model.

## Network visualization

We use the Grad-CAM method to generate heatmaps that highlight the areas of focus within the network. To verify the accuracy of the model's recognition, we compare the heatmaps generated at each stage with those produced by ConvNeXt, Swin_T, and PVT-V2-b2 at the corresponding stages. Additionally, we provide lesion annotations from expert doctors, as shown in Fig. 10. The proposed HMCFormer demonstrates a significant advantage by accurately focusing on the lesion areas at each stage, compared to ConvNeXt, Swin_T, and PVT-V2-b2.

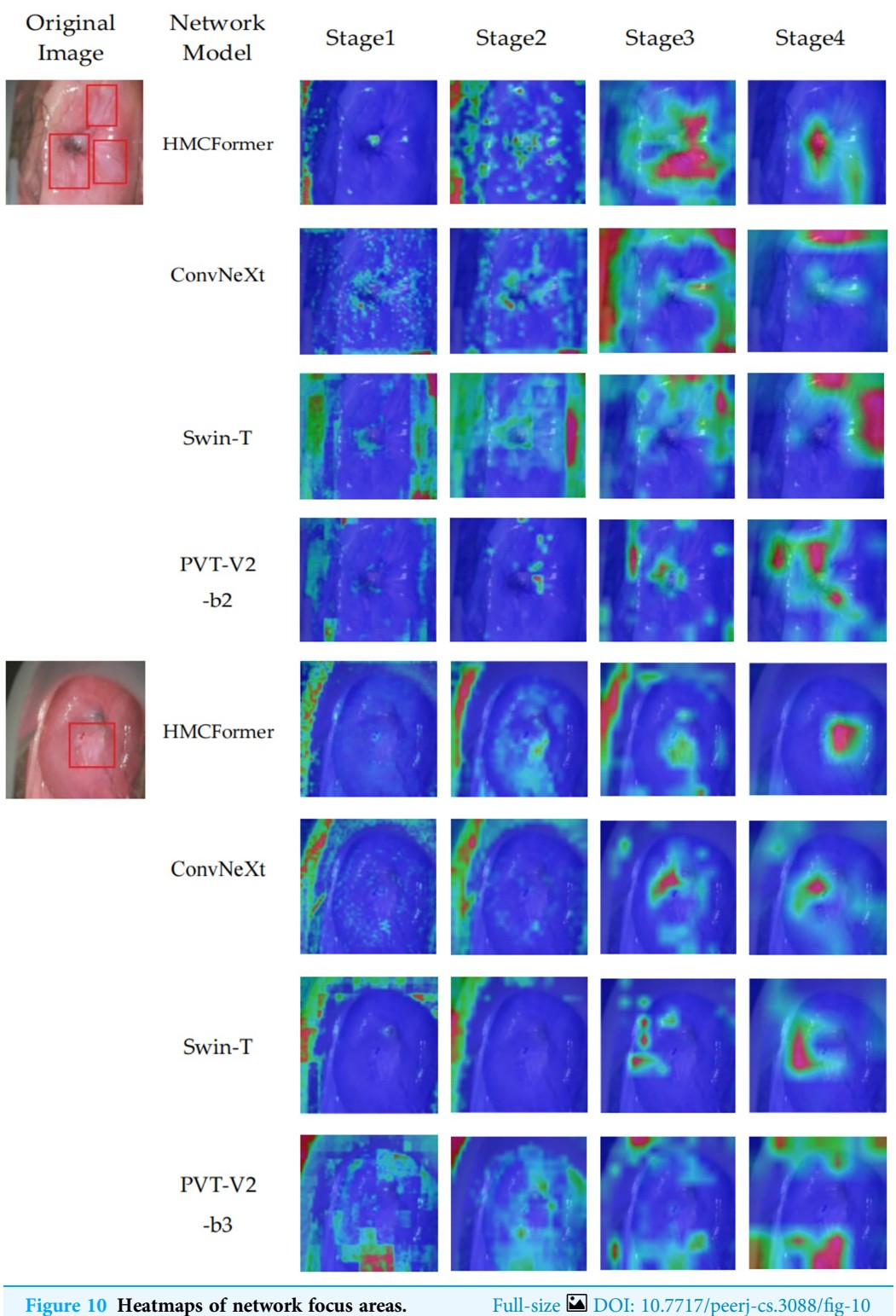

**Figure 10 Heatmaps of network focus areas.**

**Table 4 Comparison of fusion methods.**

| Fusion methods | Params (M) | FLOPs (G) | I(X,Y) | H(X) | H(Y) | H(X,Y) | Binary classification Acc | Three-class classification Acc |
|---|---|---|---|---|---|---|---|---|
| 1 | 12.1 | 1.74 | 17.97 | 10.46 | 10.24 | 2.73 | 93.4% | 90.9% |
| 2 | 12.1 | 1.75 | 8.92 | 8.30 | 13.78 | 13.17 | 95.4% | 93.7% |
| 3 | 12.1 | 1.74 | 10.50 | 10.80 | 12.84 | 13.14 | 94.7% | 92.2% |
| 4 | 13.2 | 2.08 | 9.82 | 9.03 | 12.68 | 11.89 | 93.8% | 88.7% |
| 5 | 12.7 | 1.93 | 2.69 | 6.54 | 10.02 | 13.87 | 91.6% | 89.1% |
| 6 | 12.2 | 1.77 | 9.40 | 10.32 | 12.61 | 13.53 | 96.8% | 94.1% |
| **7 (Ours)** | **12.5** | **1.78** | **9.12** | **10.72** | **12.99** | **14.59** | **97.4%** | **94.8%** |

Note:
The bold entries in the comparison table indicate the results of the model proposed in this article.

## Ablation study

### Comparison of fusion methods

In parallel CNN+Transformer networks, different fusion methods can directly affect the overall network performance. We will compare six fusion methods to identify the optimal fusion strategy for parallel CNN+Transformer networks:

1. Element-wise addition: Based on the current model in this article, modify the existing fusion method by removing all unnecessary structures, and perform fusion using element-wise addition at the end of each stage.

2. Concatenation fusion: Modify the current model by removing all unnecessary structures and concatenate the feature vectors from the CNN and Transformer branches at the end of each stage. After concatenation, use shuffle to mix the features thoroughly, then split the mixed features into two parts—one for the Transformer branch and one for the CNN branch.

3. Fusion method from *Chen et al. (2024)*: Based on the current model, modify the existing fusion method by removing all unnecessary structures and apply the fusion method from *Chen et al. (2024)*. In this method, the Transformer branch receives information from the CNN branch, but the CNN branch does not receive information from the Transformer branch.

4. Fusion method from *Yoo et al. (2023)*: Modify the current model by removing all unnecessary structures and apply the fusion method from *Yoo et al. (2023)*. At the end of each stage, first fuse the feature vectors from the CNN and Transformer branches, then add them to the feature vectors of both branches.

5. Fusion method from *Xu et al. (2024)*: Based on the current model, modify the existing fusion method by removing all unnecessary structures and apply the fusion method from *Xu et al. (2024)*. No fusion is performed at stages 0, 1, and 2; fusion occurs only at stage 3.

6. Fusion method without adaptive preprocessing: Apply the fusion method proposed in this article without the adaptive preprocessing step.

7. The fusion method proposed in this article.

**Table 5 Binary classification image enhancement ablation analysis.**

| Network model | Accuracy | Sensitivity | Specificity | Error rate | Precision | F1 |
|---|---|---|---|---|---|---|
| HMCFormer_O | 0.954 | 0.945 | 0.961 | 0.045 | 0.948 | 0.946 |
| HMCFormer_E | 0.969 | 0.966 | 0.973 | 0.030 | 0.981 | 0.974 |
| HMCFormer | 0.974 | 0.981 | 0.968 | 0.025 | 0.958 | 0.970 |

**Table 6 Ternary classification image enhancement ablation analysis.**

| Network model | Accuracy | Sensitivity | Precision | F1 |
|---|---|---|---|---|
| HMCFormer_O | 0.939 | 0.919 | 0.906 | 0.912 |
| HMCFormer_E | 0.926 | 0.893 | 0.901 | 0.897 |
| HMCFormer | 0.948 | 0.928 | 0.919 | 0.923 |

In this article, the feature vectors output at the end of stage 3 will be used to calculate H(X), H(Y), and I(X;Y). The feature vectors from the Transformer branch are denoted as X, and those from the CNN branch are denoted as Y. Table 3 presents the values of I(X,Y), H(X), and H(Y) for these seven methods at Epochs = 500.

From Table 4, Methods 1, 2, and 4 exhibit $I(X;Y) > H(X) + H(Y) - Max(H(X), H(Y))$, indicating significant redundancy in the network's feature vectors, with the highest redundancy observed in Method 1. In Methods 2, 4, and 5, the relatively low H(X) values suggest that the Transformer branch has limited feature diversity, with many similar features. Compared to all methods, the proposed method (Method 7) demonstrates the best performance across various metrics, achieving the highest H(X) value and the second-highest H(Y) value, indicating good feature diversity in both the Transformer and CNN branches. The I(X;Y) value is slightly less than $H(X) + H(Y) - Max(H(X), H(Y))$, suggesting strong synergy between the Transformer and CNN branches with low redundancy. In summary, the proposed method outperforms the other five fusion methods. Comparing Method 6 and Method 7, adaptive preprocessing effectively enhances feature fusion, leading to a notable increase in both H(X) and H(Y).

### Ablation analysis of image enhancement

The dual-color space enhancement fusion algorithm is a significant innovation in this study. To demonstrate its importance in intelligent cervical cancer identification, we proposed two validation networks: HMCFormer_O and HMCFormer_E. In HMCFormer_O, we replaced the input, which was initially the enhanced image, with the grayscale version of the original image, thereby eliminating all enhanced image inputs. In HMCFormer_E, we changed the input from the original image to the enhanced image, removing all inputs of the original image.

From Tables 5 and 6, it can be observed that in HMCFormer_O, which excludes all image augmentations, the metrics for both binary and three-class classification tasks are significantly lower than those of HMCFormer. In the binary classification task, the accuracy of HMCFormer_O is only 0.07% higher than that of the second-best TransXNet in Table 2, its sensitivity is 3.2% higher, and its F1-score is 2.7% higher than those of

TransXNet. In the three-class classification task, HMCFormer_O's accuracy is just 0.08% higher than that of the second-best TransXNet in Table 3, with its sensitivity being 1.6% higher and its F1-score 0.09% higher. These results indicate that the HMCFormer network architecture has certain advantages over existing general-purpose networks, and also confirm that the image augmentation algorithm indeed plays a crucial role.

In HMCFormer_E, where only augmented images are used without the original images, we found that its binary classification performance is excellent, achieving an accuracy of 96.9%, but still not surpassing the 97.4% accuracy of HMCFormer. In the three-class classification, the accuracy of HMCFormer_E is only 92.9%. This indicates that the image augmentation algorithm does indeed affect the grading of CIN, further validating the rationale of using original images as input for the Transformer branch and augmented images as input for the CNN branch, as well as the approach discussed in "Dual Color Space-Based Image Enhancement Technology" of converting augmented images to grayscale to remove color information.

## CONCLUSIONS

Cervical cancer is the fourth leading cause of cancer-related deaths among women worldwide. Early detection of cervical intraepithelial neoplasia can significantly increase the survival rates of patients with cervical lesions. Regular cervical cancer screening is the most effective method to reduce cervical cancer mortality. However, the current mainstream screening method, which combines TCT and HPV testing, is expensive and time-consuming. In developing countries, this cost and time burden means that most low- and middle-income women seldom choose to undergo cervical cancer screening. Although AI assistance has begun to be utilized in TCT+HPV screening—bringing certain cost savings and efficiency improvements to medical institutions—the collection, transportation, storage, and analysis of cervical cells still require professional personnel and specialized, expensive equipment. These requirements constitute the majority of the costs associated with TCT+HPV screening. Therefore, the cost savings achieved through AI assistance are almost impossible to translate into benefits for patients. AI-assisted VIA screening will, at an incredibly low cost, once again become the mainstream method for cervical cancer screening in the future. It will be widely promoted in community hospitals, rural clinics, and small- to medium-sized health examination institutions, truly becoming as routine and inexpensive a test as measuring blood pressure or blood glucose.

This article proposes a more rational CNN+Transformer fusion method that maximizes the collaborative potential between the CNN and Transformer branches. Utilizing mutual information from information theory, we demonstrate that our method effectively harnesses this collaborative potential. We introduce the HMSPE module, which enables the CNN branch to integrate multi-scale and multi-level feature extraction capabilities with minimal additional computational overhead. In our approach, the Transformer branch processes the original, unenhanced images, while the CNN branch processes images enhanced by our proposed dual color space image enhancement algorithm. This method significantly improves recognition accuracy. We have compiled and organized publicly available datasets provided by companies such as Intel and Google, obtaining

5,000 samples suitable for the VIA screening method, thereby forming the PCC5000 dataset. On this dataset, our algorithm achieves a screening accuracy of 97.4% and a grading accuracy of 94.8%.

At present, the dataset we use relies entirely on publicly available datasets from the internet. In the future, we will collaborate with multiple hospitals to continue refining our dataset. Based on the algorithm presented in this article, we will develop auxiliary diagnostic software for cervical cancer and invite gynecologists to join our platform. Suspected cases identified by our software will receive prompt secondary confirmation from professional gynecologists. Furthermore, we plan to develop an inexpensive colposcope equipped with acetic acid spraying and network transmission functions. Ultimately, cervical cancer screening will become as simple and affordable as measuring blood glucose.

### Funding
This research is supported by the National Key Research and Development Program funded project (2019YFC0117800). The funders had no role in study design, data collection and analysis, decision to publish, or preparation of the manuscript.

### Grant Disclosures
The following grant information was disclosed by the authors:
National Key Research and Development Program: 2019YFC0117800.

### Competing Interests
The authors declare that they have no competing interests.

### Author Contributions
- Bo Feng conceived and designed the experiments, performed the experiments, performed the computation work, authored or reviewed drafts of the article, and approved the final draft.
- Chao Xu analyzed the data, performed the computation work, prepared figures and/or tables, authored or reviewed drafts of the article, and approved the final draft.
- Zhengping Li conceived and designed the experiments, performed the experiments, performed the computation work, authored or reviewed drafts of the article, and approved the final draft.
- Chuanyi Zhang analyzed the data, prepared figures and/or tables, and approved the final draft.

### Data Availability
The data is available at Dryad: Feng, Bo; Xu, Chao; Li, Zhengping (2025). Cervical intraepithelial neoplasia acetic acid white images—pre-cancerous lesion three-class classification [Dataset]. Dryad. https://doi.org/10.5061/dryad.g1jwstr22.

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
