# Peer review of "HMCFormer (hierarchical multi-scale convolutional transformer): a hybrid CNN+Transformer network for intelligent VIA screening"

_PeerJ Computer Science, doi:10.7717/peerj-cs.3088_

## Round 0.1 · original submission · Major Revisions

The comments of both reviewers must be addressed in full.

·

Basic reporting

There are some shortcomings in the study, and addressing them would strengthen the work:

- Data from 2024 should be included in line 42.
- Imaging methods for cervical cancer should be discussed, and the most important ones highlighted.
- In the abstract, 'authors proposed' is used consistently; it should be replaced with 'we,' and similar corrections should be made elsewhere.
- It’s crucial to mention CNN-ViT-based studies in the literature review.
- The contributions at the end of the introduction are not presented clearly enough. They need to be articulated more academically.
- CNN-ViT-based cervical cancer studies should be included in the literature review.
- The full name of HMCFormer should be given in the abstract and introduction, with a clearer title for the model.

Experimental design

- The train-validation-test split for the dataset needs to be provided, as otherwise, the model might overfit. Common splits are 70-15-15 or 80-10-20, or similar.
- The partitioned dataset and, where possible, the number of patients should be shown in a table.
- Links to the datasets should also be presented in a table, and the links should be checked as they currently don’t work.

- If data augmentation and transfer learning were used, they should be mentioned in a few paragraphs.
- The figures are of low quality.
- The number of parameters or inference time increase when fusing models should be discussed.
- Training for only 30 epochs is insufficient, and with a batch size of 64, the GTX 2080Ti you mentioned (actually RTX 2080Ti) usually can’t handle it. It typically works with a batch size of 16-32, so adjustments are necessary.

Validity of the findings

- The number of model parameters should also be included in the ablation study section.
- Adding graphs (like accuracy and F1 scores) in the experimental section would enhance the results.
- In the discussion or a new section, limitations and future directions should be addressed.
- The discussion could be more impactful.
- The primary reasons why the model outperforms others have not been clearly stated.
- The proposed model should be compared with similar works in 2022 and beyond in a new section.

Making these adjustments would significantly strengthen the study.

Reviewer 2 ·

Basic reporting

1.The manuscript uses terms inconsistently, such as “chaotic systems” and “nonlinear systems.” Standardizing terminology throughout the text would improve clarity. Additionally, “Python” is misspelled as “Phython” in the methodology section.
2.Minor formatting inconsistencies were noted, such as inconsistent capitalization in section headings (e.g., “Methods” vs. “methodologies”). Standardizing these elements will enhance the manuscript’s professional appearance.
3.Some technical terms such as “chaotic systems” and “nonlinear dynamics” are not sufficiently defined in the introduction. Including concise definitions or examples would enhance accessibility for readers unfamiliar with these concepts.

Experimental design

4.While the search strategy aligns with Cochrane standards, the justification for excluding studies not using quantitative applications of chaos theory could be elaborated further. This would enhance transparency regarding the inclusion criteria.
5.The PRISMA flow chart in Figure 1 is informative, but its readability could be improved by enlarging text labels. Also, consider including a supplementary diagram illustrating chaotic behavior applications across medical specialties.
6.The manuscript notes that most studies used Matlab, with limited open-source availability. This is a crucial point. Expanding on the implications for reproducibility and suggesting ways to encourage open-source adoption would strengthen the discussion.

Validity of the findings

7.The use of the STRESS tool for quality assessment is appropriate, but the manuscript could benefit from a more detailed discussion of specific biases identified in the included studies. This would highlight potential gaps in the literature.
8.The manuscript effectively discusses practical applications of chaos theory. However, it could be enriched by adding a section summarizing the most promising future research directions and their potential clinical impact.

---

## Round 0.2 · Major Revisions

Thanks to the reviewers for their efforts to improve the work. However, there are still some issues that need to be addressed. Please carefully consider the comments and continue to revised the article.

Reviewer 3 ·

Basic reporting

All comments have been added in detail to the last section.

Experimental design

All comments have been added in detail to the last section.

Validity of the findings

All comments have been added in detail to the last section.

Additional comments

Review Report for PeerJ Computer Science
(HMCFormer: A Hybrid CNN+Transformer Network for Intelligent VIA Screening)

1. In this study, the Hierarchical Multi-Scale Convolutional Transformer network (HMCFormer) is proposed as a low-cost and fast solution for AI-assisted VIA screening, achieving high accuracy in cervical cancer detection.

2. In the introduction, Cervical cancer, screening methods, the importance of the subject, its relationship with artificial intelligence and the main contributions of the study to the literature are mentioned in detail and in an explanatory manner.

3. In the Related works section, transformer fusion and convolutional neural networks, procedures and criteria are mentioned. In this section, in order to highlight both the place of the subject in the literature and the differences and contributions of the study from the literature, a more in-depth comparison of convolutional neural networks and transformer fusion types should be made with the help of tables.

4. When the details of the proposed architecture are examined, it is observed that it contains a certain level of originality. However, this model has serious deficiencies in terms of comparison with the state-of-the-art models in the literature.

5. Although the dataset is separated into certain percentages as traning, validation and testing, it is recommended that the study be interpreted in detail in terms of cross-validation.

6. In order for the results to be analyzed correctly and for the proposed model to come to the fore, the evaluation metrics must be obtained completely. It is important to explain the results in depth, especially in terms of Cohen's cappa, receiver operating characteristic (ROC) curve and AUC (the area under the ROC curve) scores.

As a result, the study has the potential to make a serious contribution to the literature with the proposed artificial intelligence model in cervical cancer detection. However, the sections mentioned above should be taken into consideration.

---

## Round 0.3 · accepted · Accept

The authors have revised the paper accordingly. I believe the current version may be accepted.

·

Basic reporting

The authors have not responded to all of my questions, and some of my comments appear to have been deleted.

Experimental design

The authors have not responded to all of my questions, and some of my comments appear to have been deleted.

Validity of the findings

The authors have not responded to all of my questions, and some of my comments appear to have been deleted.

Additional comments

The authors have not responded to all of my questions, and some of my comments appear to have been deleted.

Reviewer 3 ·

Basic reporting

All comments are in the last section

Experimental design

All comments are in the last section

Validity of the findings

All comments are in the last section

Additional comments

Thanks for the revision. Both the responses to my comments and the changes in the paper are sufficient for me. Best regards.